# Safeguarding climate-resilient mangroves requires only a moderate increase in the global protected area

Alvise Dabalà [1,2] ✉, Christopher J. Brown[3], Tom Van der Stocken [4], Christina A. Buelow [5], David S. Schoeman [6,7], Daniel C. Dunn [1,2], Catherine E. Lovelock [1], Farid Dahdouh-Guebas [4,8,9,10], Jason Flower [11,12,13], Sandra Neubert [1,2,14], Kristine Camille V. Buenafe [1,2], Jason D. Everett [1,2,15,16], Kris Jypson T. Esturas [1,2,4,10] & Anthony J. Richardson [1,2,15]

Climate change and anthropogenic activities threaten biodiversity and ecosystem services. Climate-smart conservation plans address these challenges by ensuring protection of some climate-resilient areas. However, integrating climate change in the design of conservation plans is often deemed too expensive, as it may require larger networks or protecting more costly sites from a conservation perspective. Using mangroves as a case study, we evaluate the efficiency of protecting mangroves in climate-smart versus climate-naïve reserve networks. We find that climate-smart conservation plans could provide sizable benefits (13.3%) for relatively moderate increases in protected area (+7.3%). Moreover, transboundary plans, involving cooperation among countries, require less area and protect more climate-resilient mangroves than nation-by-nation plans. Implementing these strategies would improve the current protected area network for mangroves, which currently has poor climate resilience. Our methodology could potentially be tested on other ecosystems, assuming sufficient information exists regarding their distribution, biodiversity, and resilience to climate change.

The ongoing loss of biodiversity threatens the provision of critical ecosystem goods and services[1,2]. In response, conservation scientists and practitioners have developed area-based management tools, including protected areas, to safeguard species and ecosystems from anthropogenic activities[3]. Although conservation planning is used extensively to design protected areas, its application has frequently overlooked climate change[4]. In a rapidly changing world, climate-smart conservation planning—where climate risks are explicitly considered in the design of conservation plans—can produce networks more resilient to future climate change and anthropogenic activities. Climate-smart planning supports adaptation and mitigation actions, while promoting sustainable use and conservation[5]. Despite the call from the Kunming-Montreal Global Biodiversity Framework to incor-

porate climate change in conservation planning (Target 8)[6], most plans remain climate-naïve, neglecting the impacts of climate change. This is probably because of perceptions that climate-smart approaches are more complex, uncertain, data-intensive, and result in substantially more expensive conservation plans requiring larger networks or protecting sites with a higher cost for conservation (i.e., costs of acquisition, management, opportunity, transaction and damage)[7,8].

Mangroves are vital coastal habitats that provide diverse ecosystem services, such as supporting fisheries, carbon sequestration and coastal protection[9]. However, these benefits are increasingly threatened by climate change and other anthropogenic activities[10–12]. Mangroves are vulnerable to sea-level rise, especially in areas with low sediment supply and where natural or human barriers restrict their

---

landward expansion[13]. Further, intensifying drought and more frequent and severe cyclones can cause mangrove degradation and dieback[10,14]. Some examples of these events have already been reported—such as the mangrove dieback in the Gulf of Carpentaria in 2015 or in Mozambique in 2019[15,16]—but more are predicted to happen in the future[17]. The combination of climate change and the conversion of mangrove habitats for aquaculture, agriculture, and urban development is projected to accelerate mangrove loss[11]. This loss could negatively affect livelihoods[9] and release large quantities of $CO_2$[11].

Recent studies on mangroves and climate change developed models evaluating the global effects of sea-level rise[13], predicted carbon emissions by 2100 due to the loss of mangrove forests caused by anthropogenic activities and climate change[11], and projected future distributions of mangroves under climate change[18]. Despite projections of severe climate change impacts on mangroves, spatial prioritisations (i.e., the formal quantitative design of protected areas in conservation planning), have neglected climate change[19]. However, ecosystem services are likely to be lost in areas most impacted by climate change[9]. Since there is limited ability to retroactively make established protected areas more resilient to climate change, there is considerable opportunity to ensure that any newly designed protected areas maximise climate resilience[20].

Most studies designing climate-smart protected area networks have focused on climate refugia—areas projected to be climatically stable[21]. A challenging aspect of mangrove conservation is that drivers of change in mangrove cover, which can cause both losses and gains, could act differently on the landward and seaward edges of mangroves. For instance, while sea-level rise is a driver of mangrove loss on the seaward edge, it can also be a driver of mangrove gain on the landward edge through inland migration of mangroves where unimpeded by natural or anthropogenic structures[13].

Here, we prioritise areas with higher climate resilience identified using an ecological network model[22]. We quantify climate resilience on a scale from 0 (least resilient) to 100 (most resilient), with the least and most resilient corresponding to the highest and lowest probability of loss, respectively, from the model. We use this model to compare a climate-smart and climate-naïve protected area network in terms of their differences in total area and climate resilience, but meeting identical biodiversity conservation objectives. These objectives are to select a predefined percentage of the area occupied by each mangrove species in different geomorphic landscape types (here called geomorphic species—see "Methods"). We answer four key questions: (1) How much larger is a climate-smart protected area network than a climate-naïve one? (2) When targeting areas more resilient to climate change, is a global prioritisation more efficient in terms of area than one at the country scale? (3) How different are climate-smart prioritisations of the landward and seaward edges of mangroves, given their differing responses to climate change? (4) How climate-smart is the existing protected area network for mangroves compared to a climate-smart one? To our knowledge, this is the first global climate-smart conservation plan for mangroves.

## Results

### Climate-smart mangrove conservation requires only a moderate increase (7.3%) in protected area

The baseline global-scale climate-naïve prioritisation (Fig. 1a)—which identified a potential network of protected areas for mangrove plant species conservation by selecting the smallest area needed to meet all specified conservation targets—required 39.7% of the total mangrove area globally (Fig. 1c; higher areal protection targets were set for species with small distributions—see "Methods"). This network had a mean climate resilience of 38.9 (calculated as average resilience at landward and seaward edges; Fig. 1d). The climate-naïve prioritisation selected the highest percentages of mangroves in East Asia, Melanesia and South America.

By contrast, the global-scale climate-smart prioritisation (Fig. 1b)—which used a climate-smart threshold of 0.3 (i.e., selecting 30% of the most climate-resilient areas within the overall distribution of each geomorphic species that would need to be selected to reach its species-specific conservation target—see Methods)—covered 42.6% of the total mangrove area and achieved a mean climate resilience of 44.1. Thus, at the global scale and relative to the climate-naïve prioritisation, the climate-smart prioritisation had a 7.3% increase in area $\left(\frac{Area_{Climate\ smart}[42.6]-Area_{Climate\ naïve}[39.7]}{Area_{Climate\ naïve}[39.7]}\times100\%\right.$; Fig. 1c) yet had a 13.3% increase in mean climate resilience $\left(\frac{Resilience_{Climate\ smart}[44.1]-Resilience_{Climate\ naïve}[38.9]}{Resilience_{Climate\ naïve}[38.9]}\times100\%\right.$; Fig. 1d). This area increase favoured the selection in countries with more climate-resilient mangrove areas, such as the Republic of Congo, Venezuela and New Caledonia (Fig. 1b).

### A global-scale protected area network is smaller but less representative than the sum of country-scale networks

As conservation measures are generally implemented at national or local scales, we evaluated a country-scale prioritisation—i.e., a prioritisation that meets conservation targets independently in each country (sovereign dependencies and overseas territories were treated as separate countries; Fig. 2a and Supplementary Fig. 1a, b). As expected, the climate resilience for the country-scale prioritisation was greater for the climate-smart (40.2; Fig. 2d) than the climate-naïve prioritisation (37.0; Supplementary Fig. 1d). Further, compared with the global-scale prioritisations, the country-scale was less efficient in terms of total area, both for the climate-naïve $\left(\frac{Area_{Global\ scale}[39.7]-Area_{Country\ scale}[48.6]}{Area_{Global\ scale}[39.7]}\times100\%=22.4\%\right.$ larger; Supp. Figure 1c) and the climate-smart prioritisations $\left(\frac{Area_{Global\ scale}[42.6]-Area_{Country\ scale}[50.5]}{Area_{Global\ scale}[42.6]}\times100\%=18.5\%\right.$ larger; Fig. 2c).

However, in the global-scale analyses, mangroves were not selected in all countries: 46 out of 122 countries in the climate-naïve prioritisation, and 54 countries in the climate-smart prioritisation (Fig. 1c). This is because the prioritisation met its species targets in a smaller total area by selecting mangroves in places with higher species richness. For example, mangroves in the East African Coast were not selected in either global-scale prioritisation (purple bins in Fig. 1a, b). This is probably because there are no mangroves that are endemic only to Africa and because of the lower species richness in East Africa compared to the rest of the Indo-West Pacific. Hence, the prioritisation selected other areas with higher species richness where it could protect multiple species of mangroves at a lower cost (i.e., smaller area). By contrast, country-scale prioritisations had more similar percentages of mangrove areas selected across countries, evident from the lower standard deviation of the proportion of mangrove area selected by country (0.23 for country-scale climate-smart prioritisation vs 0.33 for global-scale climate-smart).

### Higher increases in resilience compared to the required additional area

We next examined how increasingly strict objectives for protecting climate-resilient mangroves would impact the additional mangrove area needed for protection. We ran climate-smart prioritisations that placed an incrementally greater emphasis on selecting climate-resilient areas within the distribution for each geomorphic species (i.e., climate-smart thresholds were increased from 0.05 to 1 at steps of 0.05—see Methods). We found that the increase in resilience relative to the area protected was proportionally greater for climate-smart thresholds <0.9 in the global-scale and <0.95 in the country-scale prioritisation (Fig. 3). Overall, for the same increase in area, the increase in resilience benefits was greater in the prioritisations that were less climate-smart than the more climate-smart prioritisations

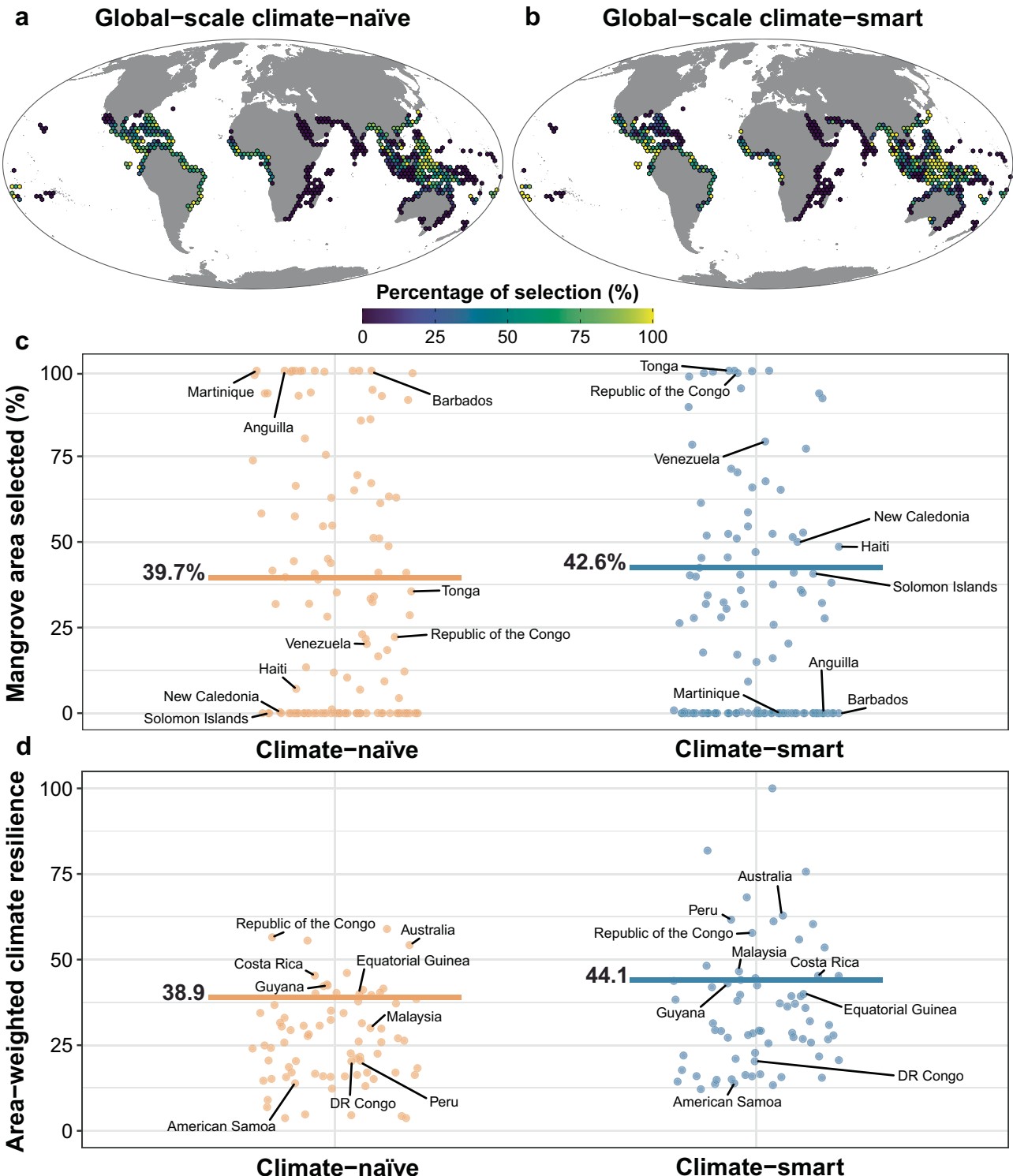

**Fig. 1 | A climate-smart prioritisation is 13.3% more resilient than a climate-naïve one for a 7.3% increase in area. a** Climate-naïve and (**b**) climate-smart global-scale spatial prioritisations. Note that the % selection is shown because the original 631 km² for each planning unit in the analysis was aggregated at a resolution of ~ 63,000 km² for visualisation. **c** % mangrove area selected and (**d**) area-weighted climate resilience of the mangrove areas selected in the prioritisations by country (each country is a dot). Both prioritisations used a climate-smart threshold of 0.3. The horizontal lines show (**c**) the total % of mangrove area selected and (**d**) the global average climate resilience.

(Fig. 3). This arises because the less climate-smart prioritisations (i.e., lower climate-smart thresholds) already include the most climate-resilient mangroves. As a result, stricter climate-smart prioritisations can expand the network only by including less climate-resilient mangrove areas.

## Different opportunities for protecting landward and seaward edges

We ran separate climate-smart prioritisations for landward and seaward edges to explore how differences in the severity and the cross-shore distribution of impacts of climate change influenced global priorities

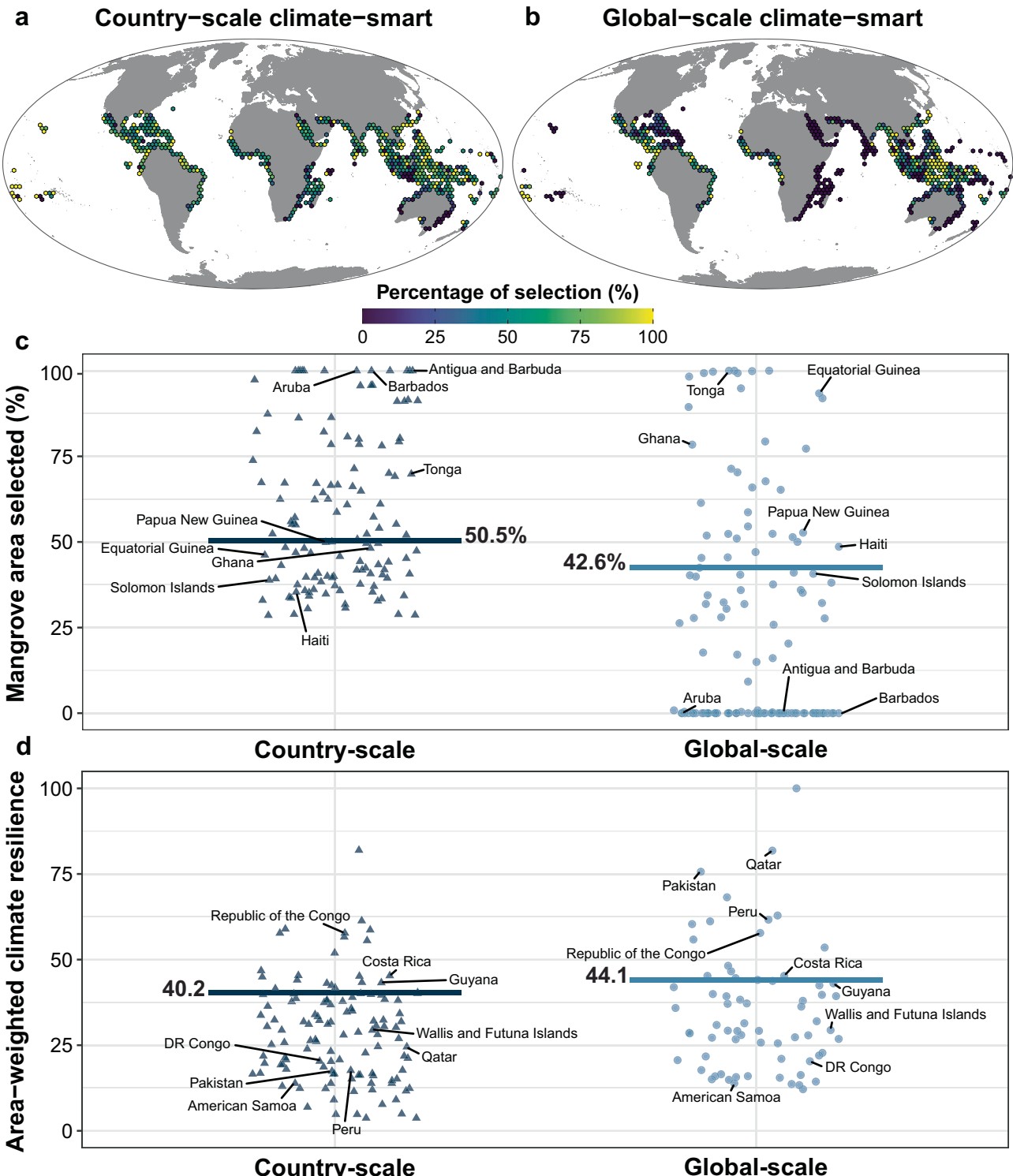

**Fig. 2 | A global-scale climate-smart prioritisation requires less area and is more resilient than a country-scale prioritisation. a** Global-scale and (**b**) country-scale climate-smart spatial prioritisation. Note that the % selection is shown because the original 631 km² for each planning unit in the analysis was aggregated at a resolution of ~ 63,000 km² for visualisation. **c** % mangrove area selected and (**d**) area-weighted climate resilience of the mangrove areas selected in the prioritisations by country (each country is a triangle or a dot). Both prioritisations used a climate-smart threshold of 0.3. The horizontal lines show (**c**) the total % of mangrove area selected and (**d**) the global average climate resilience.

(Supp. Figure 2). The global-scale climate-smart prioritisations for landward and seaward edges showed a moderate spatial agreement (Cohen's Kappa, $\varkappa = 0.38$, indicating "fair" agreement between the prioritisations). However, many countries had markedly different places selected between the two prioritisations: 57 out of 90 selected countries

ranged from "great disagreement" ($\varkappa \leq 0$; e.g., Barbados, DR Congo and Qatar) to "none to slight agreement" ($0 < \varkappa \leq 0.1$; e.g., Bangladesh, China and Colombia; Fig. 4). By contrast, there was only a small difference in the total mangrove area selected (i.e., small circles in Fig. 4) between landward and seaward edges in many countries, such as Dominican

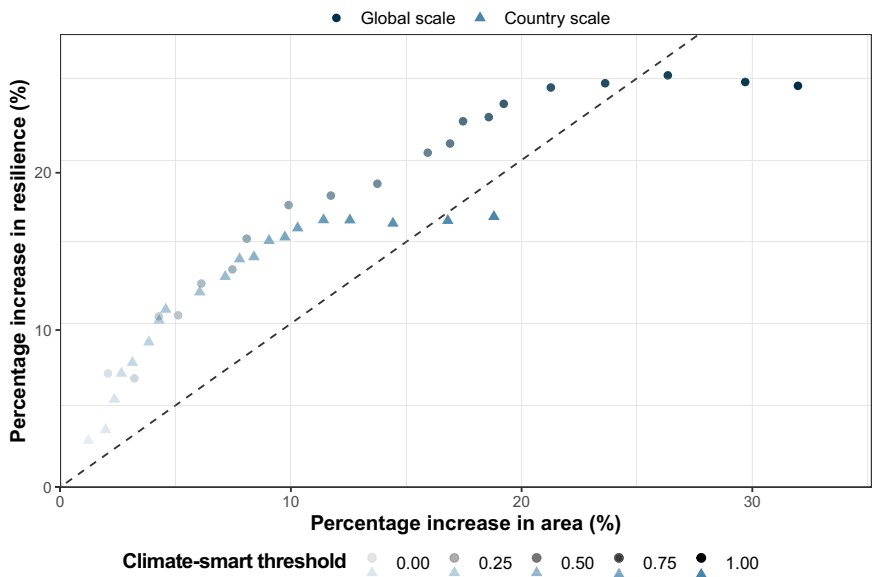

**Fig. 3 | Differences in effectiveness of global-scale vs country-scale climate-smart spatial prioritisation.** Percentage increase in area and resilience of the global-scale climate-smart prioritisation and of the country-scale climate-smart prioritisation from the respective baseline climate-naïve prioritisations for increasing climate-smart thresholds (0.05–1 with 0.05 increases—see "Methods"). The increase in climate-smart thresholds is displayed by an increase in the opacity of the dots and triangles. We added a 1:1 dashed line to aid interpretation.

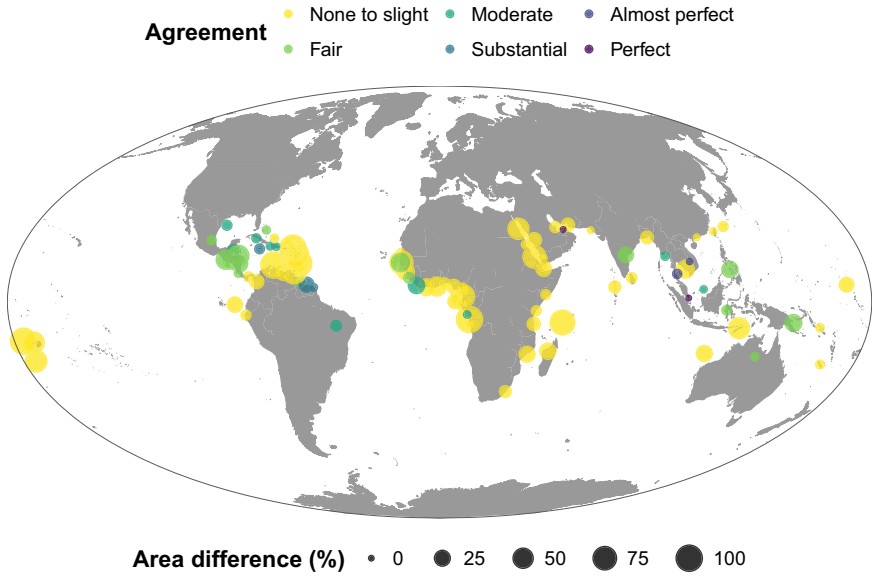

**Fig. 4 | Differences between global-scale landward and seaward prioritisations by country.** Each circle represents a country and is positioned in the centroid of the mangrove distribution in each country polygon. Circle colours represent the degree of agreement between the landward and seaward prioritisation calculated using Cohen's kappa statistic. Circle size represents the difference in the percentage of mangrove area selected between the two prioritisations (i.e., larger circles represent a greater difference in total mangrove area selected in the country). Countries in which mangroves were not selected in both prioritisations are not represented. Both landward and seaward prioritisations used a climate-smart threshold of 0.3.

Republic, French Guiana and Vietnam. However, there was a major difference in mangrove area selected in other countries (i.e., large circles in Fig. 4). For example, in Aruba, all mangroves were selected in the landward climate-smart prioritisation and no mangroves were selected in the seaward, with the opposite in Dominica.

### Poor climate-smart performance of existing protected areas compared to our climate-smart prioritisation
The current network of protected areas covers 43.1% of mangroves, similar to the total area of our global-scale climate-smart prioritisation results. We found that the current network of protected areas overlapped with 41.5% of the climate-smart selected areas. However, the current network of protected areas only met area-based conservation targets used in our prioritisations for 56 of 258 geomorphic species (102 of 258 when considering a uniform 30% target). For the geomorphic species whose targets were not met, the mean shortfall from the targets amounted to 20% of the distribution area for each. The current protected area network performs well at conserving mangroves resilient to climate change on the landward edge, as our proposed landward global-scale climate-smart prioritisation was, on

average, only 5.1% more resilient to climate change (Supplementary Fig. 3a). By contrast, the current network performs poorly at protecting the seaward edge, as our proposed seaward global-scale climate-smart network was 82.5% more resilient to climate change than the current network of protected areas (Supplementary Fig. 3b).

## Sensitivity of the results

We ran sensitivity analyses to test the robustness of our results to planning unit size (1000 km$^2$ compared with 631 km$^2$ here) and conservation targets (30% target for each geomorphic species compared to area-based targets here). We obtained comparable results when using larger planning units with greater increases in resilience than increases in area (Supplementary Fig. 4, 5), and with higher proportional increases in resilience for less climate-smart prioritisations (Supplementary Fig. 6). Even larger increases in resilience per increase in area were obtained when using uniform 30% conservation targets. Nonetheless, in this case, the proportional increase in resilience compared to the increase in area started decreasing for smaller climate-smart thresholds compared to the results using an area-based target. Country-scale prioritisations always required a larger area than the global-scale prioritisations. Also, in this case, the seaward and landward prioritisations had a "fair" agreement, but most countries showed "none to slight" agreement (Supplementary Fig. 7).

## Discussion

Effective conservation plans need to account for climate change and its future impact on species and ecosystems[4–6]. Our study demonstrates how incorporating the likely impacts of climate change into conservation plans can increase future ecosystem resilience with a relatively moderate increase in area (+7.3%). We also found that transboundary plans are both smaller and more climate-smart than nation-by-nation prioritisations. Further, our analysis revealed little agreement between climate-smart prioritisations of landward and seaward edges of mangroves because of the different effects of some of the stressors involved (e.g., subsidence and sea-level rise have a positive effect on the landward edge but a negative effect on the seaward edge, and erosion only has a negative effect on the seaward edge)[22]. This might also be the case for other coastal habitats[10]. Our study directly incorporates climate resilience for the development of area-based conservation tools for mangroves. This makes it stand out from the many mangrove studies that used modelling or satellite images to develop products that are related to climate mitigation or adaptation strategies (e.g., mangrove height and biomass[23] or high-resolution maps to monitor change[24]) but did not integrate them into planning. Although our focus was on mangrove ecosystems, our approach of developing climate-smart conservation plans could be applied to other ecosystems if data on their extent, biodiversity, and climate resilience are available.

Protected areas currently cover 43.1% of mangroves globally, meeting the 30% target (Target 3) defined by the 30×30 objective of the Kunming-Montreal Global Biodiversity Framework. However, the high percentage of mangroves in protected areas is misleading because it does not consider: (1) the global distribution of mangroves before extensive clearing and conversion[25]; (2) uncertainties in tidal boundaries that influence whether mangroves are included within protected areas[26]; and (3) the effectiveness of protected areas in reducing mangrove losses[27]. In fact, anthropogenic influences are a continuing (albeit decelerating) driver of mangrove loss globally[12] and considerable human-driven losses are occurring within protected areas[27]. In response, organisations are articulating more ambitious targets for mangrove conservation. For example, the Global Mangrove Alliance has called for mangrove protected area coverage to be doubled from 40% to 80% of their current distribution[28]. We found that the current network does not protect the most climate-resilient mangroves, especially along their seaward edge. It also fails to achieve most

of the area-based and of the uniform conservation targets for mangrove plant species we used in our analysis (see Methods), despite covering a similar amount of area to our climate-smart prioritisation. Overall, our results highlight opportunities to expand the current protected area network to better represent mangrove diversity and simultaneously increase resilience to climate change.

Previous studies emphasise that transboundary cooperation can reduce conservation costs[29–32], although they did not explicitly consider climate change. Our results show that in designing climate-smart protected areas, planning at a global scale not only requires less area but also selects more climate-resilient areas compared to planning at a local scale. However, global-scale prioritisations have their limitations. For example, they do not ensure the representativity of species diversity in different areas or provide sufficient replication to reduce extinction risk[33]. Further, developing global-scale prioritisations could exacerbate the conservation burden for countries in the Global South[34]. While country-scale planning could solve some of these issues, it is more costly and could still miss other aspects of conservation. For example, the distribution of most species transcends borders, requiring transboundary management[35]. Moreover, climate-smart planning is challenging at a local scale, given that climate model outputs are currently appropriate for use at global to regional scales[36]. There are likely to be advantages of multi-scale conservation planning, not only considering the benefits of transboundary cooperation but also including the value of incorporating local-scale variability and replication. Developing large-scale prioritisations for mangroves and other systems helps identify priorities that could inform investments and actions, but need to be adapted to local circumstances, allowing adaptive adjustment during implementation[37]. Multiple protected areas that cross transnational boundaries have been developed in terrestrial and marine ecosystems, such as the Sangha Tri-national parks in the Congo basin (i.e., Lobéké National Park in Cameroon, Dzanga-Ndoki National Park in the Central African Republic and Nouabalé-Ndoki National Park in the Republic of Congo)[38], the Kavango Zambezi Transfrontier Conservation Area[39] and the Eastern Tropical Pacific Marine Corridor between Colombia, Costa Rica, Ecuador and Panama[40]. In mangrove systems, the Togo-Benin mono transboundary reserve in West Africa has led to positive outcomes[41], and a joint conservation initiative meant to be valuable for mangrove conservation between Kenya and Tanzania is underway[42,43]. These examples underscore the value of strengthening international cooperation in conservation while ensuring that global or regional efforts translate into effective local-level conservation actions.

The probability of future net loss of mangroves under climate change is typically higher for the seaward than the landward mangrove edge[22], resulting in more opportunities to prioritise the protection of climate-resilient mangroves on seaward edges. Considering the dissimilarity we found between landward and seaward prioritisations for individual countries, there is a need to develop strategies and actions to protect mangrove plant species specific to landward and seaward forest edges. For instance, landward migration can be facilitated by removing barriers whilst seaward progradation by increasing or trapping sediment. Both landward and seaward resilience can also be increased by actively planting propagules as inundation increases, or improving ecological connectivity[22]. These actions could be especially helpful in coastal cities threatened by flooding caused by sea-level rise and land subsidence, promoting the implementation of ecosystem-based defences[44]. However, on the landward edge, mangroves may replace other land uses and could lead to declines in freshwater biodiversity[45,46], making landward actions potentially less attractive than seaward ones. Setting aside areas to accommodate future landward migration through legal agreements could help maintain productivity for landholders while ensuring future mangrove migration[47]. Climate change could also drive shifts in species composition[45,48]. For example, on the seaward edge, protected areas that now safeguard

mangroves could be important for seagrass biodiversity in the future, shifting the provisioning of ecosystem services[48]. Considering the response of coastal ecosystems to climate change, together with the development of land-sea management strategies (e.g., the 'ridge-to-reef' approach[49]), could improve future conservation efforts. Moreover, tailoring conservation actions to seaward and landward dynamics might also be valuable for other coastal ecosystems, such as saltmarshes, that expand or contract at different edges of their distributions, similar to mangroves[50]. Overall, developing strategies for conservation actions that consider the different effects of climate change on landward and seaward sides could improve the future conservation of mangroves.

Our analysis has several caveats. First, the IUCN species distribution maps provide only broad distribution ranges[51]. As in Dabalà et al.[52], we improved these low-resolution species distributions by intersecting them with the latest high-resolution mangrove distributions from Global Mangrove Watch[53]. More accurate distribution maps would improve our results, but given the urgency of the threats, we need to proceed using available data. Second, we quantified the future resilience of mangroves to climate change using a relatively complex network model that requires capacity and time to develop. We selected this model as it evaluates the response of mangroves to interacting processes known to cause substantial mangrove loss or gain under climate change beyond just sea-level rise (e.g., intense storms, extreme rainfall and flooding), and allowed us to examine landward and seaward edges. If such climate impact models are not available, we suggest using simpler climate-smart approaches, such as climate metrics[21]. Third, the network model projects climate resilience of entire mangrove forests rather than individual species and therefore does not capture fine-scale variability related to species-specific responses to climate change. Nonetheless, we included all plant species in mangroves in the analysis following the IUCN Red List of Threatened Species[51]. Growing mostly in the landward periphery of mangrove habitats and extending into terrestrial marginal zones, some of these species (in some cases defined as mangrove associates) could have different migration patterns and climate resilience than other mangrove species. Hence, our analysis is not directly applicable to the implementation of on-the-ground local conservation plans, which require finer-scale data and analysis. Fourth, we used a single future emission scenario, SSP5-8.5, because this is the only scenario available with the model of mangrove climate change. Using the "worst-case" scenario aligns with our focus on the cost of including climate change in conservation, and more optimistic scenarios would result in lower costs. Moreover, using mid-term projections (years 2040–2060) as we did here minimises the differences between emission scenarios compared to longer-term projections[36]. However, when developing on-the-ground conservation networks, we recommend using the full range of future scenarios[36]. Fifth, there are large uncertainties in climate projections, particularly for some variables such as future cyclone tracks. We used the most accurate data available wherever possible. Sixth, multiple alternatives are available to define conservation targets (e.g., uniform targets, area-based targets, targets based on the threat status of the species)[54]. While the area-based targets we used are somewhat arbitrary, they are some of the most widely adopted in conservation planning (e.g., Hanson et al.[55], Claes et al.[56]). Nonetheless, we observed similar patterns, with even higher increases in resilience for moderate increases in area, when using uniform targets (i.e., 30%; Supplementary Figs. 4, 5). Other approaches might provide different results in detail but are unlikely to change the main findings. Seventh, additional tests confirm that the identification of specific conservation priorities was sensitive to input uncertainties. Nonetheless, our overall conclusions proved robust. Eight, although connectivity is one of the key principles of conservation planning, we did not consider it to avoid including a possible confounding factor in the analysis[57]. Nonetheless, we acknowledge the importance of incorporating it for developing robust on-the-ground conservation plans[58]. Last, to simplify the analysis, we did not consider the opportunity cost related to the implementation and management of protected areas for mangroves. However, ecosystem services provided by mangrove conservation could mitigate these costs[52]. Further, we did not consider the cost of future land acquisition for protecting mangroves migrating landward, which could be explored in future analyses.

Biodiversity plays a critical role in mitigating climate change through carbon sequestration, facilitating adaptation to climate change by safeguarding coastal settlements from storm surges, and providing other valuable ecosystem services, including clean air, water and food[9,59]. These benefits are compromised by biodiversity loss, driven by anthropogenic activities and exacerbated by climate change[1]. Developing and implementing climate-smart conservation ensures the future sustainability of these benefits and safeguards biodiversity, including those depending upon critical habitats such as mangroves[5]. Our study demonstrates that a moderate increase in the extent of protected areas can result in a much more climate-smart protected area network. Our results can inform decision-makers, highlighting opportunities for establishing national and international conservation strategies to develop effective climate-smart conservation actions.

## Methods
### Study area
The study area, mangrove biodiversity, and conservation features were prepared following the approach of Dabalà et al.[52] for developing large-scale conservation plans for mangroves. However, we have used updated databases for mangrove distribution and biophysical typology, incorporated data on projected effects of climate change on mangroves[22], and did not include information on ecosystem services as they were not our focus. The study area (planning region) was defined as the global extent of mangroves, based on the 2022 version of the Global Mangrove Watch (GMW) v3.1.4 dataset (https://www.globalmangrovewatch.org)[53]. This area was subdivided into 7559 hexagonal planning units, each with a spatial resolution of 631 km$^2$ (~ 0.25° alongshore) and defined under the Mollweide equal-area coordinate reference system (ESRI:54009). All spatial datasets were reprojected to match this resolution and coordinate system before analysis. Country boundary data was sourced from the *rnaturalearth* package[60]–separately including sovereign dependencies and overseas territories–to aggregate the results by continent and run a country-scale analysis. We performed all analyses using the R statistical computing environment version 4.4.0[61].

### Mangrove biodiversity
The spatial distribution of mangrove biodiversity came from two datasets. First, the geographic range data for the 65 most common mangrove plant species were obtained from the IUCN Red List of Threatened Species[51] (Supplementary Note 1), and overlapped with the planning units. Non-intersecting planning units were assigned species that intersected with the nearest planning unit. Second, mangrove biophysical typology data v3.0 from Worthington et al.[62] were included to account for geomorphic variability. These data categorise mangroves into geomorphic classes (i.e., deltaic, estuarine, lagoonal and open-coast mangroves) based on the influence of geomorphic and ecological processes on mangrove forest structure and associated taxa[62]. To refine biodiversity data, species ranges were further subdivided by biophysical typology classes, effectively creating separate conservation features (i.e., mangrove "geomorphic species" feature) as input data for the analysis. The mangrove area covered by each geomorphic species feature in a planning unit was calculated as the area covered by the specific biophysical typology used to define that geomorphic species.

## Conservation features

To ensure the protection of mangrove plant species, the distribution of the different mangrove species was used to set minimum area requirements for each species within each mangrove biophysical typology. Area-weighted conservation targets were calculated for each species using a $\log_{10}$-interpolation, following Rodrigues et al. [63], ranging from a maximum target of 100% for species with ranges smaller than 10,000 km² to a minimum target of 10% for species with ranges exceeding 250,000 km². The target obtained for a mangrove species was used for each of its geomorphic subgroups.

## Climate change resilience

The inclusion of information on resilience to climate change allowed prioritisation to maximise resilience. The probability of mangrove net gain/stability under future climate change estimated by Buelow et al. [22] was used as a metric for mangrove climate resilience. This value was derived from a simulation of a qualitative network model that predicts a binary outcome of either net gain/stability or net loss of the seaward and landward edges of mangroves in response to projected climatic and anthropogenic pressures under emission scenario SSP5-8.5 for the period 2040–2060. The landward edge was qualitatively defined in the network model as a mangrove forest that can migrate landwards and was classified as such if it intersects with land, whereas the seaward edge can prograde seawards and was classified as such if it intersects with the ocean[22]. The probability of future net gain/stability of mangroves was estimated through simulations wherein the relative strength of the interactions between pressures and seaward/landward mangroves was randomly parameterised. A probability of 100% means that 100% of the network model simulations resulted in net gain/stability of mangroves and 0% resulted in loss. Conversely, a probability of 0% means that 0% of the network model simulations resulted in net gain/stability of mangroves and 100% resulted in loss. The probability of seaward and landward net gain/stability was estimated for each mangrove forest unit ($n = 3983$) of the mangrove biophysical typology map[62], which were intersected with our 0.25° hexagonal planning units to obtain a climate resilience value for each planning unit. The analysis was run both using a mean value of climate resilience for seaward and landward edges, and separately using either the seaward or landward value.

## Existing protected areas

Data on protected areas were obtained from the World Database on Protected Areas[64] and processed for analysis using the *wdpar* R package[65]. Only protected areas classified as "Designated", "Inscribed" or "Established" were included, while United Nations Educational, Scientific and Cultural Organisation (UNESCO) Biosphere Reserves and point localities lacking spatial extent were excluded. The percentage of mangroves covered by protected areas in each planning unit was calculated by intersecting the biophysical typology map with the protected areas to calculate the area of each mangrove geomorphic species to be protected in each planning unit. The effectiveness of the existing protected area network was assessed by evaluating the number of conservation targets reached, its climate resilience, and its overlap with our resulting networks of climate-smart protected areas[64].

## Spatial prioritisation

Climate-smart networks for area-based management were identified using spatial prioritisations through the R package *prioritizr*[66]. All the prioritisations were generated using the minimum-set objective function, which aims to select areas that reach all the conservation targets at the minimum cost. They were solved using the Gurobi solver (v11.0.3)[67], with an optimality gap of 0.01%. To ensure the same selection priority for each km² of mangrove area, the area of mangroves in each planning unit was used as the cost for the selection, with each prioritisation aiming to reach all conservation targets while minimising this cost. A climate-naïve prioritisation was generated without incorporating data on climate resilience. A second prioritisation was generated—for a mean value of climate resilience and also separately for a value of climate resilience of the seaward and landward mangrove edges—by incorporating data on climate resilience (i.e., making the designs climate-smart) using the climate-priority area method from Buenafe et al. [21]. This method splits the distribution of each geomorphic species into: (1) climate-priority areas, defined as the most climate-resilient areas within the geomorphic species' distribution, and (2) the rest of the distribution. Climate-priority areas were identified as areas whose climate resilience exceeded a given percentile for each geomorphic species' resilience values globally. Climate-priority areas were assigned a conservation target of 100%. Non-climate priority areas were then to reach the overall target for each of the geomorphic species (e.g., to select a total of 30% of the distribution of the species). To adjust the method to our species-specific conservation targets, the threshold used to select the climate-priority area of a geomorphic species was rescaled based on the conservation target of the geomorphic species. This meant that a threshold of 0.05 selects as a climate-priority area 5% of the overall distribution of the geomorphic species that would need to be selected to reach its conservation target. In a sensitivity analysis, we ran climate-smart prioritisations using incremental thresholds for the selection of the climate-priority areas (i.e., from 5% to 100% of the conservation target, with increments of 5%). The same climate-naïve and climate-smart prioritisations were then generated using the geomorphic species of mangroves split by countries. This provides more freedom to the algorithm to select areas that are more resilient to climate change.

## Sensitivity analysis

We ran two separate sensitivity analyses for each prioritisation using different-sized planning units (i.e., 1000 km²) and uniform 30% targets (as opposed to area-based targets) based on the Kunming-Montreal Global Biodiversity Framework Target 2 of protecting 30% of the land and the sea by 2030[6].

## Reporting summary

Further information on research design is available in the Nature Portfolio Reporting Summary linked to this article.

# Data availability

The datasets used in our study are publicly available online. These include: (1) the global map of mangroves is available for download from the Global Mangrove Watch[53] website (https://www.globalmangrovewatch.org/); (2) IUCN distribution of mangrove species[51] is available at: https://www.iucnredlist.org/resources/spatial-data-download; (3) the global biophysical mangrove typology[62] is available for download from a Zenodo repository: [https://doi.org/10.5281/zenodo.8340259][68]; and (4) the data on the probability of future mangrove loss (i.e., mangrove climate resilience) is available from a Zenodo repository: [https://doi.org/10.5281/zenodo.13668500] The datasets generated during the current study are available at ref. 69: [https://doi.org/10.5281/zenodo.17728764].

# Code availability

Code to run the analysis in this study is currently available at: https://github.com/AlviDab/Mangroves_ClimateChange. The code is archived in a Zenodo digital repository[69]: [https://doi.org/10.5281/zenodo.17728764].

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

## Acknowledgements

F.D.G. and K.J.T.E. were supported by the Erasmus Mundus Joint Master Degree in Tropical Biodiversity and Ecosystems – TROPIMUNDO, which is funded by the European Commission (EC contract N° 2019-1451). C.J.B. was supported by a Future Fellowship (FT210100792) from the Australian Research Council. J.F. received funding from the Waitt Institute.

## Author contributions

Conceptualisation, A.D., C.J.B, T.V.d.S, C.A.B., D.S.S., D.C.D., C.E.L., F.D.G., J.F. and A.J.R.; Methodology, A.D., C.J.B, T.V.d.S, C.A.B., D.S.S., D.C.D. and A.R.; Formal analysis, A.D. and J.E.; Writing – original draft, A.D.; Writing – review & editing, A.D., C.J.B., T.V.d.S, C.A.B., D.S.S., D.C.D., C.E.L., F.D.G., J.F., S.N., K.C.V.B., K.J.T.E. and A.J.R.

## Competing interests

Christopher J. Brown co-leads the Global Mangrove Alliance's Science Working Group, an organisation dedicated to advancing mangrove conservation. No other authors have any competing interests.

## Additional information

[1]School of the Environment, The University of Queensland, Brisbane, Australia. [2]Centre for Biodiversity and Conservation Science (CBCS), The University of Queensland, St Lucia, QLD, Australia. [3]Institute for Marine and Antarctic Studies, University of Tasmania, Taroona, Tasmania, Australia. [4]Ecology, Evolution & Genetics (bDIV) research group, Biology Department, Vrije Universiteit Brussel, VUB, Pleinlaan, Brussel, Belgium. [5]Thriving Oceans Research Hub, School of Geosciences, The University of Sydney, Sydney, Australia. [6]Ocean Futures Research Cluster, School of Science and Engineering, University of the Sunshine Coast, Maroochydore, Qld, Australia. [7]Department of Zoology, Centre for African Conservation Ecology, Nelson Mandela University, Gqeberha, South Africa.

⁸Systems Ecology and Resource Management Unit, Department of Biology of Organisms, Université Libre de Bruxelles - ULB, Av. F.D. Roosevelt 50, CPi 264/1, Brussels, Belgium. ⁹Mangrove Specialist Group (MSG), Species Survival Commission (SSC), International Union for the Conservation of Nature (IUCN), Zoological Society of London, London, UK. ¹⁰Interfaculty Institute of Social-Ecological Transitions, Université Libre de Bruxelles - ULB, Av. F.D. Roosevelt 50, Brussels, Belgium. ¹¹Marine Science Institute, University of California Santa Barbara, Santa Barbara, CA, USA. ¹²Bren School of Environmental Science & Management, University of California Santa Barbara, Santa Barbara, CA, USA. ¹³Environmental Market Solutions Lab, University of California Santa Barbara, Santa Barbara, CA, USA. ¹⁴Centre for Ecology and Conservation, Faculty of Environment, Science and Economy, University of Exeter, Penryn, UK. ¹⁵CSIRO Environment, Queensland BioSciences Precinct (QBP), St Lucia, Australia. ¹⁶Centre for Marine Science and Innovation, School of Biological, Earth and Environmental Science, The University of New South Wales, Sydney, Australia. ✉e-mail: a.dabala@uq.edu.au

