## [Peer Review File · Nature Communications]

Safeguarding climate-resilient mangroves requires only a moderate increase in the global protected area

Corresponding Author: Mr Alvise Dabalà

Version 0:

Reviewer comments:

Reviewer #1

(Remarks to the Author)

In this article, the authors compare two types of mangrove conservation plans: those that are naive to the future impacts of climate change ("climate-naive") and those that incorporate a measure of climate resiliency into the plan ("climate-smart"). The measure of climate resiliency was estimated using a network model developed by Buelow et al., while the conservation plans were created using a spatial prioritisation algorithm proposed by Buenafe et al. The authors provide several descriptive results from their analysis. Broadly, the major results include:

1. Using the climate-smart prioritisation resulted in a relatively small increase (7.3%) of total protected area compared to the climate-naive approach; however, the climate-smart conservation plan led to a 13.3% increase in mean climate resilience.
2. Country-scale naive and climate-smart plans had similar total measures of mean climate resilience as their global counterparts, however, the percent of protected area under country-specific plans was larger than global.
3. The prioritisation of landward or seaward edges resulted in large differences in country-level areas selected for conservation for some parts of the globe.
4. The current network of protected area has some overlap with climate-smart prioritizations, however, it does not meet many of the conservation targets considered in the climate-smart prioritization.

I found the article to be interesting; I hope that the following comments are helpful.

Comments:

1. There are a lot of user-specified choices (or parameters) that went into the chosen spatial prioritisation algorithm. For example, (i) the specified conservation targets, (ii) the choice of metric to use for climate resilience, (iii) the threshold used to select climate-priority areas, (iv) the choice of hexagonal planning unit size and location, etc. Given that the results are largely descriptive (and that there is no obvious approach for uncertainty quantification), I'm left wondering how sensitive these results are to some of the aforementioned choices. The sensitivity study shown in Fig. 3 is a good start – would it be possible to assess how results change if (a) the grid resolution is changed or (b) conservation targets are perturbed?
2. On a related note: The caption for figure 3 states that the climate-smart thresholds are increased from 0.05 to 1, however, in the Methods section (lines 393–394) it mentions that you only consider "[...] from 5% to 30% of the conservation target."
3. On line 338 you mention that conservation targets are calculated as in Rodrigues et al., and yet in the discussion (line 234) you mention they are somewhat arbitrary. Are there any alternatives that might be considered (and if so, again, how sensitive are the results are to this choice)?
4. In general, I found the figures to nicely complement the text. The one exception is Figure 5. I question if this figure is really needed, or if the critical information (i.e., the relative similarity in the landward distributions vs. the difference in seaward distributions) might be better conveyed by box plots rather than kernel density plots.

References:

- Buelow et al. (2024) "Uncertainties in forecasts of ecosystem persistence under climate change."
- Buenafe et al. (2023) "A metric-based framework for climate-smart conservation planning."
- Rodrigues et al. (2004) "Global Gap Analysis: Priority Regions for Expanding the Global Protected-Area Network."

(Remarks on code availability)

Reviewer #2

(Remarks to the Author)

Review report

The study is focused on safeguarding climate-resilient mangroves requires a small increase in the global protected area. It aims to identify and delineate climate smart and climate naïve mangrove areas based on a prioritisation analysis using the biophysical conditions at a global and country scale. The rationale is to inform and improve conservation planning for mangrove forest ecosystems in a time of growing climate change, where climate resilient ecosystems need protection. The study is timely, the results are significant and informative, and are followed through to the discussion. Readers will easily recognize the scope of the study but might get loss in understanding the analytical framework and ecological analysis due to its complexity and bias towards some mangrove-rich countries and regions.

Title

I think the title could be revised to: "Safeguarding climate-resilient mangroves requires an increase in the global network of protected areas". I think "... small increase in global PA" is vague in itself because the word small was not quantified in spatial extent. But connecting mangrove PA across regions and countries (in a continuum) will improve of mangrove ecosystem connectivity (reduce mangrove fragmentation), increase biodiversity richness, facilitate migration of wildlife along such corridors/linkages and build more climate resilience.

Introduction

Line 48: Area-based management tools for biodiversity conservation are not only developed by conservation scientists, conservation practitioners play a crucial role.

The idea in line 60-61 on mangrove ecosystem services is repeated in line 68-70

The idea of climate smart PA across transnational boundaries has been long implemented in terrestrial forest ecosystem conservation planning in areas of rich/hotspots biodiversity, where a network of PA were created to ensure their climate resilience and strengthened transboundary conservation such as the Sangha Tri-national parks in the Congo basin. These include the Lobéké National Park in Cameroon, Dzanga-Ndoki National Park in the Central African Republic and Nouabalé-Ndoki National Park in the Republic of Congo. Do we have similar initiatives for mangrove forest ecosystems, and how was this exemplified?

Line 97-98: I think that the last research question should be the first, as it sets the stage for the second.

Methods

Line 311: what is slight differences methodology?

Line 313: Please provide the appropriate citation in the end, Bunting et al. (2022)?

Line 342: Was the climate resilience for each species or dominant species (geomorphic species) considered during analysis for landward and seaward edge effects?

Climate change resilience and spatial prioritisation

I think that the authors should have presented some case studies to demonstrate even deeper how the ecological model captured significant differences in biophysical conditions between where mangroves are found to grow. For example: *Avicennia marina* grow in the arid and hyper-arid countries in the Middle East (Saudi Arabia, Qatar, UAE, Bahrain and even Kuwait) as dwarf mangroves of less than 6 meters tall in some areas, mostly in small patches. How was the climate priority and climate naïve areas decided for mangroves in the Middle East, considering their unique climate, biophysical and biogeographical conditions? Was the ecological model able to determine the climate resilience of mangroves in this region based on the biophysical condition and selected the geomorphic species as a regional planning unit? The coastal land boundaries in these countries are mostly developed and landward expansion of mangroves might be limited, does this suggest that such areas are non-climate resilient or climate naïve? That further means the probability of landward net gain/stability for each mangrove forest unit in the region was 0? The inherent climate condition of the region may easily be considered as climate naïve areas even though mangroves have survived there for more than 30 years, even with coastal development being a major deterrent of landward mangrove expansion. The countries in this region should be considered in country-scale prioritization.

Mangroves (especially *Avicennia germinans* and *Rhizophora* species) grow in semi-arid conditions in the Sine Saloum and Cassamance delta of Senegal. The Sine Saloum and Cassamance mangrove areas that were degraded by the droughts in the 1980s and human activities are now bare mudflats; were they considered as climate naïve areas? What is the probability that such areas could be restored and mangroves thrive again with both opportunities for landward and seaward expansion based on the model? This country should be considered in country-scale prioritization.

Results

Line 100-101: Again, I think that the climate resilience of mangrove forest ecosystems can be improved by increasing connectivity and continuum between them through creating PA, not by increasing small patches here and there.

Line 137: More explanation is needed for why mangroves were not selected in all countries in the global-scale analyses.

Line 139: Why were the mangroves in the African East Coast not selected in either global-scale climate naïve and climate

smart prioritization? Mangroves in Mozambique's Sofala and Zambezi districts are prone to cyclone extreme weather conditions and periodic droughts, such that their climate resilience merits some consideration in this study; they represent the largest continuum of mangrove cover in southern Africa. How can the climate resilience of climate naïve areas be improved such that more mangrove areas are regained and conserved?

Discussion

I think that the discussion could benefit from an improved structure, wherein, having two subsections focused on the two key results, as this will make it easy to read and create a more focused understanding of the key ideas under discussion. Also, adding clear subsections for recommendations and challenges/limitations for future conservation planning and climate resilient mangrove forest ecosystems at different spatial scales.

Line 211. What is negligible additional cost? How do we quantify that? A statistical analysis should be completed to support that rather than using the word negligible.

Line 212: Transboundary conservation plans would increase mangrove network connectivity and build more climate resilience. That is why increasing mangrove connectivity is a better way to improving climate resilience than simply increasing small areas of mangrove.

Line 304: "... moderate increase in the extent of protected areas...": In my opinion this statement lacks quantitative clarity needed at this level as I mentioned earlier. Earlier the authors used the word "small increase" and here "moderate increase", which is a clear weakness of the study in terms of uncertainty on what the authors really mean. Increasing and maintaining mangrove network connectivity translates to increasing mangrove area and conserving PA for improved climate resilience.

(Remarks on code availability)

Reviewer #3

(Remarks to the Author)

Title: Safeguarding climate-resilient mangroves requires a small increase in the global protected area

This manuscript addresses an important and timely topic; however, it requires revisions to enhance its quality.

Abstract:

The authors stated: "Integrating climate change into the design of conservation plans is often considered too expensive, as it may require larger networks or the protection of more costly sites."

To improve clarity, the authors should specify which coastal cities they are referring to—whether global examples or those from a specific region—in the abstract.

Introduction:

This section is among the weakest in the document. Except for the third paragraph, the other paragraphs are overly generalized, lacking specific examples related to mangroves, climate change, and coastal vulnerability. It's essential to update this section by incorporating these suggestions.

Methodology:

While the methodology is clear and well-presented, the authors need to highlight the innovation behind their approach. Additionally, they should clarify why they chose a method similar to that of Dabalà et al. (2023), providing a justification for this choice.

Results:

This section is quite strong.

However, if the authors emphasize that coastal cities, particularly mega coastal cities, require more focus on climate resilience and mangrove protection, it would significantly enhance the article.

While Figures 1 and 2 (a and b) are unclear. Please provide high-resolution, larger images.

Discussion:

It is good; however, it needs specific examples when mentioning local scale. Comparing and contrasting similar works is also suggested for the revisions.

Some of the examples to read and cite:

Sunkur, R., Kantamaneni, K., Bokhoree, C., Rathnayake, U., & Fernando, M. (2024). Mangrove mapping and monitoring using remote sensing techniques towards climate change resilience. *Scientific Reports*, 14(1), 6949.

Simard, M., Fatoyinbo, L., Thomas, N. M., Stovall, A. E., Parra, A., Barenblitt, A., ... & Hajnsek, I. (2025). A New Global Mangrove Height Map with a 12 meter spatial resolution. *Scientific data*, 12(1), 15.

(Remarks on code availability)

I conducted a review of some sections of the code, but unfortunately, I was unable to access the complete codebase as my laptop was not made available. This limited access hinders my ability to provide a thorough assessment.

Reviewer #4

(Remarks to the Author)

Conclusions contradicting the evidence: The topic, as well as the methodology of the study, are exclusively based on mangrove ecosystems. Therefore, the conclusions/claims made from the study should only be about mangroves (but not other ecosystems). However, in the abstract, the authors are “generalizing” the findings in to other ecosystems. For example, the last sentence of the abstract says “These findings could also be applied in other ecosystems” where this statement is not supported by the data collection/analysis/ findings of the study. The methodology of the study is largely based “sea-level rise” and hence the findings are not applicable for the ecosystems that are not affected by sea-level rise. When it says “other ecosystems” they may include even terrestrial ecosystems which are not affected by sea-level rise. Such contradicting sentences of the abstract of the manuscript needs to be revised focusing the mangrove ecosystems only. (However, in the Discussion section the sentence-“Although our focus was on mangrove ecosystems, our approach of developing climate-smart conservation plans could be applied to other ecosystems if data on their extent, biodiversity, and climate resilience is available” is fine).

Methodology: When it says “edge” (seaward and landward mangrove edges), does the “edge” included a “single line” (the forest boundary line) or “a forest transect” ? . If it was a forest transect, how much was the width of that transect?

Under the mangrove plant list (As shown in the In the supplementary materials), “true mangrove species” as well as “mangrove associates” (e.g. *Acanthus ebracteatus*, *Acanthus volubilis*, *Acrostichum aureum*, *Acrostichum danaeifolium*, *Acrostichum speciosum*, *Dolichandrone spathacea*, *Phoenix paludosa*) have been pooled together as “mangrove plant species”. The “mangrove associates” are not true mangrove species and they are terrestrial plant species co-growing with mangroves. Being originally terrestrial plants, with the sea-level rise, those mangrove associates would easily migrate in to landward (if space is available. Thus, their migration patterns as well as resilience levels would be different from true mangrove species. We suggest that author should check if this point would be a concern during their analysis.

Discussion: Need to elaborate/ name what are those “different stressors”.

Citations: need to provide citations for the “, the IUCN species distribution maps”.

(Remarks on code availability)

Version 1:

Reviewer comments:

Reviewer #1

(Remarks to the Author)

Thank you for responding to my initial comments. I am satisfied with the revision.

(Remarks on code availability)

Reviewer #2

(Remarks to the Author)

I am generally satisfied with the responses to my initial comments. The authors have provided greater inclusion of geographical areas and clarity, which has improved the global appeal of the results. The figures are concise and informative and communicate greatly the results.

However, I still think that there is much uncertainty in the results, evident by the way that the changes in the prioritisation of geographical area and species richness can change the results.

For example, the topic (and in the abstract) moves from “... a small increase ...” as this “Safeguarding climate-resilient mangroves requires a small increase in the global protected area” to a “... a moderate increase” as this “Safeguarding climate-resilient mangroves requires only a moderate expansion in the global protected area”: This suggest that the main

result has changed following the revision; why this change, any explanation for this? What scale was used to define “+7.3%” as “relatively moderate” expansion in protected area”?

From this change, it is clear that changes in the prioritisation selection for climate-naïve and climate smart species and geographical areas influences the results on the area of mangrove required to improve climate resilience of mangroves (ecosystems). Following this line of thought, I would suggest that the authors add a paragraph on the limitation of the methodology used (if any).

“Moderate increase” is used in the abstract while moderate expansion is used in the title; both have been used interchangeably, but do the words mean the same in the context and scope of this study? Do the authors use “increases” to mean increase in the area cover of PAs by creating new PAs and expansion to mean a growth in the size or areal extent of existing PAs? Some clarity my might help.

This minor comments do not undermine the quality of the work and its timely importance in conservation practice and decision making at larger spatial scales. I admire very much the inclusion of the sentence that the model is not applicable at local scale decision-making on mangrove conservation, this is very important to note.

(Remarks on code availability)

Reviewer #3

(Remarks to the Author)

Authors significantly revised the manuscript.
All of my comments have been addressed.

(Remarks on code availability)

Technical Problem of my computer.

Reviewer #4

(Remarks to the Author)

Authors have sufficiently addressed the review points shown by me. I have no further suggestions on the revised manuscript.

(Remarks on code availability)

Authors have sufficiently addressed the review points shown by me. I have no further suggestions on the revised manuscript.

Replies to reviewers are in blue text.

REVIEWER COMMENTS

Reviewer #1 (Remarks to the Author):

In this article, the authors compare two types of mangrove conservation plans: those that are naive to the future impacts of climate change (“climate-naive”) and those that incorporate a measure of climate resiliency into the plan (“climate-smart”). The measure of climate resiliency was estimated using a network model developed by Buelow et al., while the conservation plans were created using a spatial prioritisation algorithm proposed by Buenafe et al. The authors provide several descriptive results from their analysis. Broadly, the major results include:

1. Using the climate-smart prioritisation resulted in a relatively small increase (7.3%) of total protected area compared to the climate-naive approach; however, the climate-smart conservation plan led to a 13.3% increase in mean climate resilience.
2. Country-scale naive and climate-smart plans had similar total measures of mean climate resilience as their global counterparts, however, the percent of protected area under country-specific plans was larger than global.
3. The prioritisation of landward or seaward edges resulted in large differences in country-level areas selected for conservation for some parts of the globe.
4. The current network of protected area has some overlap with climate-smart prioritizations, however, it does not meet many of the conservation targets considered in the climate-smart prioritization.

I found the article to be interesting; I hope that the following comments are helpful.

Thank you for your positive feedback!

Comments:

1. There are a lot of user-specified choices (or parameters) that went into the chosen spatial prioritisation algorithm. For example, (i) the specified conservation targets, (ii) the choice of metric to use for climate resilience, (iii) the threshold used to select climate-priority areas, (iv) the choice of hexagonal planning unit size and location, etc. Given that the results are largely descriptive (and that there is no obvious approach for uncertainty quantification), I’m left wondering how sensitive these results are to some of the aforementioned choices. The sensitivity study shown in Fig. 3 is a good start – would it be possible to assess how results change if (a) the grid resolution is changed or (b) conservation targets are perturbed?

Changed. Thank you for pointing this out. We ran additional sensitivity analyses using a 1000 km² grid resolution and uniform conservation targets (see Supplementary Fig. 4-7). We refer to these supplementary results in the main text (L 218-226), highlighting that the sensitivity analyses were consistent with the study's current main messages. Results

reported in the main text are conservative, as we obtained even higher increases in resilience for smaller increases in area when using uniform 30% targets.

More specifically, the changes to the main text are (L 218-226): “We obtained comparable results when using larger planning units with greater increases in resilience than increases in area (Supp. Fig. 4-5), and with higher proportional increases in resilience for less climate-smart prioritisations (Supp. Fig. 6). Even larger increases in resilience per increase in area were obtained when using uniform 30% conservation targets. Nonetheless, in this case, the proportional increase in resilience compared to the increase in area started decreasing for smaller climate-smart thresholds compared to the results using an area-based target. Country-scale prioritisations always required a larger area than the global-scale prioritisations. Also, in this case, the seaward and landward prioritisations had a “fair” agreement, but most countries showed “none to slight” agreement (Supp. Fig. 7).”

The additions to the Supplementary Material are:

Supp. Fig. 4 | Percentage of total mangrove area selected by the prioritisations used for sensitivity analysis. Bar plots reporting the percentage of mangrove area selected by the climate-smart and climate-naïve, country-scale and global-scale prioritisations produced for the sensitivity analysis. These prioritisations used larger planning units than those of the baseline prioritisations (1000 km²) or uniform 30% conservation targets for each mangrove geomorphic species.

Supp. Fig. 5 | Average area-weighted climate resilience of mangroves selected by the prioritisations used for sensitivity analysis. Plots reporting the average climate resilience of mangrove area selected by the climate-smart and climate-naïve, country-scale and global-scale prioritisations produced for the sensitivity analysis. These prioritisations used larger planning units than those of the baseline prioritisations (1000 km²) or uniform 30% conservation targets for each mangrove geomorphic species.

Supp. Fig. 6 | Sensitivity analysis of the differences in effectiveness of global-scale vs country-scale climate-smart spatial prioritisation. Percentage increase in area and resilience of the global-scale climate-smart prioritisation and of the country-scale climate-smart prioritisation from the respective baseline climate-naïve prioritisations, increasing climate-smart thresholds (0.05–1 with 0.05 increases—see Methods). These prioritisations used **a**, uniform 30% conservation targets for each geomorphic species, or **b**, larger planning units (1000 km²) than those of the baseline prioritisations. The increase in climate-smart thresholds is displayed by an increase in the opacity of the dots and triangles. We added a 1:1 dashed line to aid interpretation.

a

Uniform conservation targets (30%)

b

Planning units of 1000 km²

Agreement

● None to slight	● Moderate	● Almost perfect
● Fair	● Substantial	● Perfect

Area difference % • 0 ● 25 ● 50 ● 75 ● 100

Supp. Fig. 7 | Sensitivity analysis of the differences between global-scale landward and seaward prioritisations by country. Each circle represents a country and is positioned in the centroid of the mangrove distribution in each country polygon. Circle colours represent the degree of agreement between the landward and seaward prioritisation calculated using Cohen's kappa statistic. Circle size represents the difference in the percentage of mangrove area selected between the two prioritisations (i.e., larger circles represent greater difference in total mangrove area selected in the country). These prioritisations used **a**, uniform 30% conservation targets for each geomorphic species, or **b**, larger planning units (1000 km²) than those of the baseline prioritisations. Countries in which mangroves were not selected in both prioritisations are not represented. Both landward and seaward prioritisations used a climate-smart threshold of 0.3.

We also expanded on the use of the targets in the caveats section (L 336-342):

“Sixth, multiple alternatives are available to define conservation targets (e.g., uniform targets, area-based targets, targets based on the threat status of the species)¹. While the area-based targets we used are somewhat arbitrary, they are some of the most widely adopted in conservation planning (e.g., Hanson et al.², Claes et al.³). Nonetheless, we

observed similar patterns, with even higher increases in resilience for moderate increases in area, when using uniform targets (i.e., 30%; Supp. Fig. 4-5). Other approaches might provide different results in detail but are unlikely to change the main findings.”

We also included that we ran these sensitivity analyses in the Methods section (L 457-460):

“We ran two separate sensitivity analyses for each prioritisation using different sized planning units (i.e., 1000 km²) and uniform 30% targets (as opposed to area-based targets) based on the Kunming-Montreal Global Biodiversity Framework Target 2 of protecting 30% of the land and the sea by 2030⁴.”

2. On a related note: The caption for figure 3 states that the climate-smart thresholds are increased from 0.05 to 1, however, in the Methods section (lines 393–394) it mentions that you only consider “[...] from 5% to 30% of the conservation target.”

Changed. Thank you for highlighting the mistake (L 451-453):

“In a sensitivity analysis, we ran climate-smart prioritisations using incremental thresholds for the selection of the climate-priority areas (i.e., from 5% to 100% of the conservation target, with increments of 5%).”

3. On line 338 you mention that conservation targets are calculated as in Rodrigues et al., and yet in the discussion (line 234) you mention they are somewhat arbitrary. Are there any alternatives that might be considered (and if so, again, how sensitive are the results are to this choice)?

Changed. The uncertainty associated with the alternative ways that targets can be calculated is now described in the Caveats section (L 336-342):

“Sixth, multiple alternatives are available to define conservation targets (e.g., uniform targets, area-based targets, targets based on the threat status of the species)¹. While the area-based targets we used are somewhat arbitrary, they are some of the most widely adopted in conservation planning (e.g., Hanson et al.², Claes et al.³). Nonetheless, we observed similar patterns, with even higher increases in resilience for moderate increases in area, when using uniform targets (i.e. 30%; Supp. Fig. 4-5). Other approaches might provide different results in detail but are unlikely to change the main findings.”

4. In general, I found the figures to nicely complement the text. The one exception is Figure 5. I question if this figure is really needed, or if the critical information (i.e., the relative similarity in the landward distributions vs. the difference in seaward distributions) might be better conveyed by box plots rather than kernel density plots.

Changed. We have moved the figure to the supplementary material. Instead, we now just report the numeric result in the text.

References:

- Buelow et al. (2024) “Uncertainties in forecasts of ecosystem persistence under climate change.”
 - Buenafe et al. (2023) “A metric-based framework for climate-smart conservation planning.”
 - Rodrigues et al. (2004) “Global Gap Analysis: Priority Regions for Expanding the Global Protected-Area Network.”
-

Reviewer #2 (Remarks to the Author):

Review report

The study is focused on safeguarding climate-resilient mangroves requires a small increase in the global protected area. It aims to identify and delineate climate smart and climate naïve mangrove areas based on a prioritisation analysis using the biophysical conditions at a global and country scale. The rationale is to inform and improve conservation planning for mangrove forest ecosystems in a time of growing climate change, where climate resilient ecosystems need protection. The study is timely, the results are significant and informative, and are followed through to the discussion. Readers will easily recognize the scope of the study but might get loss in understanding the analytical framework and ecological analysis due to its complexity and bias towards some mangrove-rich countries and regions.

Thank you for your positive comments!

Title

I think the title could be revised to: “Safeguarding climate-resilient mangroves requires an increase in the global network of protected areas”. I think “... small increase in global PA” is vague in itself because the word small was not quantified in spatial extent. But connecting mangrove PA across regions and countries (in a continuum) will improve of mangrove ecosystem connectivity (reduce mangrove fragmentation), increase biodiversity richness, facilitate migration of wildlife along such corridors/linkages and build more climate resilience.

Changed. We have now changed “increase” to “expansion” to highlight that we are not advocating for developing small protected areas (L 1-2):

The title is now: “Safeguarding climate-resilient mangroves requires only a moderate expansion in the global protected area”.

Introduction

Line 48: Area-based management tools for biodiversity conservation are not only developed by conservation scientists, conservation practitioners play a crucial role.

Changed. Good point! We included conservation practitioners in the sentence (L 49-51):

“The ongoing loss of biodiversity threatens the provision of critical ecosystem goods and services^{5,6}. In response, conservation scientists and practitioners have developed area-based management tools, including protected areas, to safeguard species and ecosystems from anthropogenic activities⁷.”

The idea in line 60-61 on mangrove ecosystem services is repeated in line 68-70

Changed. We removed lines 68-70.

The idea of climate smart PA across transnational boundaries has been long implemented in terrestrial forest ecosystem conservation planning in areas of rich/hotspots biodiversity, where a network of PA were created to ensure their climate resilience and strengthened transboundary conservation such as the Sangha Tri-national parks in the Congo basin. These include the Lobéké National Park in Cameroon, Dzanga-Ndoki National Park in the Central African Republic and Nouabalé-Ndoki National Park in the Republic of Congo. Do we have similar initiatives for mangrove forest ecosystems, and how was this exemplified?

Changed. We have included some examples (L 273-285):

“Developing large-scale prioritisations for mangroves and other systems helps identify priorities that could inform investments and actions, but need to be adapted to local circumstances, allowing adaptive adjustment during implementation⁸. Multiple protected areas that cross transnational boundaries have been developed in terrestrial and marine ecosystems, such as the Sangha Tri-national parks in the Congo basin (i.e. Lobéké National Park in Cameroon, Dzanga-Ndoki National Park in the Central African Republic and Nouabalé-Ndoki National Park in the Republic of Congo)⁹, the Kavango Zambezi Transfrontier Conservation Area¹⁰ and the Eastern Tropical Pacific Marine Corridor between Colombia, Costa Rica, Ecuador and Panama¹¹. In mangrove systems, the Togo-Benin mono transboundary reserve in West Africa has led to positive outcomes¹², and a joint conservation initiative meant to be valuable for mangrove conservation between Kenya and Tanzania is underway^{13,14}. These examples underscore the value of strengthening international cooperation in conservation while ensuring that global or regional efforts translate into effective local-level conservation actions.”

Line 97-98: I think that the last research question should be the first, as it sets the stage for the second.

Thanks, we thought it was a good idea and tried revising the aims and results so that the last research question came first. However, we opted to keep the questions in the original order because we compare the current network of protected areas with the results of our prioritisations in the second to last section of the results (“Poor climate-smart performance of existing protected areas compared to our climate-smart prioritisation”). Moving this section to the beginning would require a description of the prioritisation before presenting their results, possibly confusing the readers.

Methods

Line 311: what is slight differences methodology?

Changed. We have now made the differences clear (L 363-367):

“The study area, mangrove biodiversity, and conservation features were prepared following the approach of Dabalà et al (2023)¹⁵ for developing large-scale conservation plans for mangroves. However, we have used updated databases for mangrove distribution and biophysical typology, incorporated data on projected effects of climate change on

mangroves¹⁶, and did not include information on ecosystem services as they were not our focus.”

Line 313: Please provide the appropriate citation in the end, Bunting et al. (2022)?

Thank you. We have included the correct reference in the end (L 367-369):

“The study area (planning region) was defined as the global extent of mangroves, based on the 2022 version of the Global Mangrove Watch (GMW) v3.1.4 dataset (<https://www.globalmangrovetwatch.org>)¹⁷.”

Line 342: Was the climate resilience for each species or dominant species (geomorphic species) considered during analysis for landward and seaward edge effects?

Changed. No, the network model does not have species-level resolution. We now include this in the caveats (L 314-328):

“Second, we quantified the future resilience of mangroves to climate change using a relatively complex network model that requires capacity and time to develop. We selected this model as it evaluates the response of mangroves to interacting processes known to cause substantial mangrove loss or gain under climate change beyond just sea-level rise (e.g., intense storms, extreme rainfall and flooding), and allowed us to examine landward and seaward edges. If such climate impact models are not available, we suggest using simpler climate-smart approaches, such as climate metrics¹⁸. Third, the network model projects climate resilience of entire mangrove forests rather than individual species and therefore does not capture fine-scale variability related to species-specific responses to climate change. Nonetheless, we included all plant species in mangroves in the analysis following the IUCN Red List of Threatened Species¹⁹. Growing mostly in the landward periphery of mangrove habitats and extending into terrestrial marginal zones, some of these species (in some cases defined as mangrove associates) could have different migration patterns and climate resilience than other mangrove species. Hence, our analysis is not directly applicable to the implementation of on-the-ground local conservation plans, which requires finer-scale data and analysis.”

Climate change resilience and spatial prioritisation

I think that the authors should have presented some case studies to demonstrate even deeper how the ecological model captured significant differences in biophysical conditions between where mangroves are found to grow. For example:

Avicennia marina grow in the arid and hyper-arid countries in the Middle East (Saudi Arabia, Qatar, UAE, Bahrain and even Kuwait) as dwarf mangroves of less than 6 meters tall in some areas, mostly in small patches. How was the climate priority and climate naïve areas decided for mangroves in the Middle East, considering their unique climate, biophysical and biogeographical conditions? Was the ecological model able to determine the climate resilience of mangroves in this region based on the biophysical condition and selected the geomorphic species as a regional planning unit? The coastal land boundaries in these countries are mostly developed and landward expansion of mangroves might be limited, does this suggest that such areas are non-climate resilient or climate naïve? That further

means the probability of landward net gain/stability for each mangrove forest unit in the region was 0? The inherent climate condition of the region may easily be considered as climate naïve areas even though mangroves have survived there for more than 30 years, even with coastal development being a major deterrent of landward mangrove expansion. The countries in this region should be considered in country-scale prioritization.

Mangroves (especially *Avicennia germinans* and *Rhizophora* species) grow in semi-arid conditions in the Sine Saloum and Cassamance delta of Senegal. The Sine Saloum and Cassamance mangrove areas that were degraded by the droughts in the 1980s and human activities are now bare mudflats; were they considered as climate naïve areas? What is the probability that such areas could be restored and mangroves thrive again with both opportunities for landward and seaward expansion based on the model? This country should be considered in country-scale prioritization.

Changed. We have included in the caveats that the resilience model is not species-specific, that it might not capture fine-scale variability, and that it should not be used for local decision making (L 314-328):

“Second, we quantified the future resilience of mangroves to climate change using a relatively complex network model that requires capacity and time to develop. We selected this model as it evaluates the response of mangroves to interacting processes known to cause substantial mangrove loss or gain under climate change beyond just sea-level rise (e.g., intense storms, extreme rainfall and flooding), and allowed us to examine landward and seaward edges. If such climate impact models are not available, we suggest using simpler climate-smart approaches, such as climate metrics¹⁸. Third, the network model projects climate resilience of entire mangrove forests rather than individual species and therefore does not capture fine-scale variability related to species-specific responses to climate change. Nonetheless, we included all plant species in mangroves in the analysis following the IUCN Red List of Threatened Species¹⁹. Growing mostly in the landward periphery of mangrove habitats and extending into terrestrial marginal zones, some of these species (in some cases defined as mangrove associates) could have different migration patterns and climate resilience than other mangrove species. Hence, our analysis is not directly applicable to the implementation of on-the-ground local conservation plans, which requires finer-scale data and analysis.”

Results

Line 100-101: Again, I think that the climate resilience of mangrove forest ecosystems can be improved by increasing connectivity and continuum between them through creating PA, not by increasing small patches here and there.

Changed. We have now changed “increase” to “expansion” to highlight that we are not advocating for developing small protected areas (L 104-105):

“Climate-smart mangrove conservation requires only a moderate expansion in protected area”

We also incorporated into the caveats section that connectivity was not incorporated, but that it is important to consider it when developing on-the-ground conservation plans (L 342-345):

“Seventh, although connectivity is one of the key principles of conservation planning, we did not consider it to avoid including a possible confounding factor in the analysis²⁰. Nonetheless, we highlight the importance of incorporating it for developing on-the-ground conservation plans²¹.”

Line 137: More explanation is needed for why mangroves were not selected in all countries in the global-scale analyses.

Changed (L 141-144):

“However, in the global-scale analyses, mangroves were not selected in all countries: 46 out of 122 countries in the climate-naïve prioritisation, and 54 countries in the climate-smart prioritisation (Fig. 1c). This is because the prioritisation met its species targets in a smaller total area by selecting mangroves in places with higher species richness.”

Line 139: Why were the mangroves in the African East Coast not selected in either global-scale climate naïve and climate smart prioritization? Mangroves in Mozambique’s Sofala and Zambezi districts are prone to cyclone extreme weather conditions and periodic droughts, such that their climate resilience merits some consideration in this study; they represent the largest continuum of mangrove cover in southern Africa. How can the climate resilience of climate naïve areas be improved such that more mangrove areas are regained and conserved?

Changed (L 144-149):

“For example, mangroves in the African East Coast were not selected in either global-scale prioritisation (purple bins in Fig. 1a-b). This is probably because there are no mangroves that are endemic only to Africa and because of the lower species richness in East Africa compared to the rest of the Indo-West Pacific. Hence, the prioritisation selected other areas with higher species richness where it could protect multiple species of mangroves at lower cost (i.e., smaller area).”

Discussion

I think that the discussion could benefit from an improved structure, wherein, having two subsections focused on the two key results, as this will make it easy to read and create a more focused understanding of the key ideas under discussion. Also, adding clear subsections for recommendations and challenges/limitations for future conservation planning and climate resilient mangrove forest ecosystems at different spatial scales.

Changed. We updated the discussion to improve the narrative. More specifically, we report the focus of each paragraph in the discussion.

- Summary of the main results from our analysis

- Opportunities for improving and expanding the current protected area network by incorporating climate change
- Trade-offs and synergies between transboundary and local-scale conservation planning
- Opportunities for differential seaward and landward conservation
- Challenges and limitations
- Conclusions

We would have added headings to make the discussion subsections clearer, but it is against Nature Communications policy.

Line 211. What is negligible additional cost? How do we quantify that? A statistical analysis should be completed to support that rather than using the word negligible.

Changed (L 229-231):

“Our study demonstrates how incorporating the likely impacts of climate change into conservation plans can increase future ecosystem resilience with a relatively moderate increase in area (+7.3%).”

Line 212: Transboundary conservation plans would increase mangrove network connectivity and build more climate resilience. That is why increasing mangrove connectivity is a better way to improving climate resilience than simply increasing small areas of mangrove.

Changed. We have now made it clearer that we are not advocating for developing small protected areas for mangroves. We have also included a paragraph on the importance of incorporating connectivity in the development of conservation plans for mangroves (L 342-345):

“Seventh, although connectivity is one of the key principles of conservation planning, we did not consider it to avoid including a possible confounding factor in the analysis²⁰. Nonetheless, we acknowledge the importance of incorporating it for developing robust on-the-ground conservation plans²¹.”

Line 304: “... moderate increase in the extent of protected areas...”: In my opinion this statement lacks quantitative clarity needed at this level as I mentioned earlier. Earlier the authors used the word “small increase” and here “moderate increase”, which is a clear weakness of the study in terms of uncertainty on what the authors really mean. Increasing and maintaining mangrove network connectivity translates to increasing mangrove area and conserving PA for improved climate resilience.

Changed. We now report the percentage increase required in the abstract (L 40-42) and use moderate throughout the text.

“We found that climate-smart conservation plans could provide sizable benefits for only a relatively moderate increase in protected area (+7.3%).”

Reviewer #3 (Remarks to the Author):

Title: Safeguarding climate-resilient mangroves requires a small increase in the global protected area

This manuscript addresses an important and timely topic; however, it requires revisions to enhance its quality.

Thank you for your positive feedback!

Abstract

The authors stated: "Integrating climate change into the design of conservation plans is often considered too expensive, as it may require larger networks or the protection of more costly sites."

To improve clarity, the authors should specify which coastal cities they are referring to—whether global examples or those from a specific region—in the abstract.

Changed. We clarify that we refer to “costly” from a conservation perspective (L 37-39):

“However, integrating climate change in the design of conservation plans is often deemed too expensive, as it may require larger networks or protecting more costly sites from a conservation perspective.”

We also expanded on the cost of conservation in the introduction and added more references (L 59-62):

“This is probably because of perceptions that climate-smart approaches are more complex, uncertain, data-intensive, and result in substantially more expensive conservation plans requiring larger networks or protecting sites with a higher cost for conservation (i.e., costs of acquisition, management, opportunity, transaction and damage)^{22,23}.”

Introduction

This section is among the weakest in the document. Except for the third paragraph, the other paragraphs are overly generalized, lacking specific examples related to mangroves, climate change, and coastal vulnerability. It's essential to update this section by incorporating these suggestions.

Thank you. We have now included specific examples in the introduction relative to mangroves, climate change and coastal vulnerability (L 67-72):

“Further, intensifying drought and more frequent and severe cyclones can cause mangrove degradation and dieback^{24,25}. Some examples of these events have already been reported—such as the mangrove dieback in the Gulf of Carpentaria in 2015 or in Mozambique in 2019^{26,27}—but more are predicted to happen in the future²⁸. The combination of climate

change and the conversion of mangrove habitats for aquaculture, agriculture, and urban development is projected to accelerate mangrove loss²⁹.”

Methodology

While the methodology is clear and well-presented, the authors need to highlight the innovation behind their approach. Additionally, they should clarify why they chose a method similar to that of Dabalà et al. (2023), providing a justification for this choice.

Changed. We highlighted the innovation in our study (L 101-102):

“To our knowledge, this is the first global climate-smart conservation plan for mangroves.”

We also clarified why we prepared the layers following Dabalà et al. (2023) (L 363-367):

“The study area, mangrove biodiversity, and conservation features were prepared following the approach of Dabalà et al (2023)¹⁵ for developing large-scale conservation plans for mangroves. However, we have used updated databases for mangrove distribution and biophysical typology, incorporated data on projected effects of climate change on mangroves¹⁶, and did not include information on ecosystem services as they were not the focus here.”

Results

This section is quite strong.

However, if the authors emphasize that coastal cities, particularly mega coastal cities, require more focus on climate resilience and mangrove protection, it would significantly enhance the article.

Changed, thank you. We have included more information on the value of mangrove conservation and on focusing on climate resilience in coastal cities in the Discussion (L 291-298):

“For instance, landward migration can be facilitated by removing barriers whilst seaward progradation by increasing or trapping sediment. Both landward and seaward resilience can also be increased by actively planting propagules as inundation increases, or improving ecological connectivity¹⁶. These actions could be especially helpful in coastal cities threatened by flooding caused by sea-level rise and land subsidence, promoting the implementation of ecosystem-based defences³⁰. However, on the landward edge, mangroves may replace other land uses and could lead to declines in freshwater biodiversity^{31,32}, making landward actions potentially less attractive than seaward ones.”

While Figures 1 and 2 (a and b) are unclear. Please provide high-resolution, larger images.

Changed. We have now provided high-resolution images.

Discussion

It is good; however, it needs specific examples when mentioning local scale. Comparing and contrasting similar works is also suggested for the revisions.

Some of the examples to read and cite:

Sunkur, R., Kantamaneni, K., Bokhoree, C., Rathnayake, U., & Fernando, M. (2024). Mangrove mapping and monitoring using remote sensing techniques towards climate change resilience. *Scientific Reports*, 14(1), 6949.

Simard, M., Fatoyinbo, L., Thomas, N. M., Stovall, A. E., Parra, A., Barenblitt, A., ... & Hajnsek, I. (2025). A New Global Mangrove Height Map with a 12 meter spatial resolution. *Scientific data*, 12(1), 15.

Changed. Thank you.

We added comparisons with similar work (L 236-241):

“Our study is the first that directly incorporates climate resilience into the development of area-based conservation tools for mangroves. This sets it apart from the many mangrove studies that used modelling or satellite images to develop products that are related to climate mitigation or adaptation strategies (e.g., mangrove height and biomass³³ or high-resolution maps to monitor change³⁴) but did not integrate them into planning.”

Reviewer #3 (Remarks on code availability):

I conducted a review of some sections of the code, but unfortunately, I was unable to access the complete codebase as my laptop was not made available. This limited access hinders my ability to provide a thorough assessment.

Changed. We have updated the code and it is accessible at:

https://github.com/AlviDab/Mangroves_ClimateChange

A demo of the code that uses a smaller dataset to make it less computationally intensive is available at: https://github.com/AlviDab/Mangroves_ClimateChange/tree/demo

Reviewer #4 (Remarks to the Author):

Conclusions contradicting the evidence: The topic, as well as the methodology of the study, are exclusively based on mangrove ecosystems. Therefore, the conclusions/claims made from the study should only be about mangroves (but not other ecosystems). However, in the abstract, the authors are “generalizing” the findings in to other ecosystems. For example, the last sentence of the abstract says “These findings could also be applied in other

ecosystems” where this statement is not supported by the data collection/analysis/ findings of the study. The methodology of the study is largely based “sea-level rise” and hence the findings are not applicable for the ecosystems that are not affected by sea-level rise. When it says “other ecosystems” they may include even terrestrial ecosystems which are not affected by sea-level rise. Such contradicting sentences of the abstract of the manuscript needs to be revised focusing the mangrove ecosystems only.(However, in the Discussion section the sentence-“Although our focus was on mangrove ecosystems, our approach of developing climate-smart conservation plans could be applied to other ecosystems if data on their extent, biodiversity, and climate resilience is available” is fare).

Changed. Thank you for pointing this out (L 45-47):

“Our methodology could potentially be tested on other ecosystems, assuming sufficient information exists regarding their distribution, biodiversity, and resilience to climate change.”

Methodology: When it says “edge” (seaward and landward mangrove edges), does the “edge” included a “single line” (the forest boundary line) or “a forest transect” ? . If it was a forest transect, how much was the width of that transect?

Changed. We now clarify the meaning of ‘edge’ in the manuscript (L 404-407):

“The landward edge was qualitatively defined in the network model as mangrove forest that can migrate landwards and was classified as such if it intersects with land, whereas the seaward edge can prograde seawards and was classified as such if it intersects with the ocean¹⁶.”

Under the mangrove plant list (As shown in the In the supplementary materials), “true mangrove species” as well as “mangrove associates” (e.g. *Acanthus ebracteatus*, *Acanthus volubilis*, *Acrostichum aureum*, *Acrostichum danaeifolium*, *Acrostichum speciosum*, *Dolichandrone spathacea*, *Phoenix paludosa*) have been pooled together as “mangrove plant species”. The “mangrove associates” are not true mangrove species and they are terrestrial plant species co-growing with mangroves. Being originally terrestrial plants, with the sea-level rise, those mangrove associates would easily migrate in to landward (if space is available. Thus, their migration patterns as well as resilience levels would be different from true mangrove species. We suggest that author should check if this point would be a concern during their analysis.

Changed. We have included this point in the caveats section (L 320-328):

“Third, the network model projects climate resilience of entire mangrove forests rather than individual species and therefore does not capture fine-scale variability related to species-specific responses to climate change. Nonetheless, we included all plant species in mangroves in the analysis following the IUCN Red List of Threatened Species¹⁹. Growing mostly in the landward periphery of mangrove habitats and extending into terrestrial marginal zones, some of these species (in some cases defined as mangrove associates), could have different migration patterns and climate resilience than other mangrove species. Hence, our analysis is not directly applicable to the implementation of on-the-ground local conservation plans, which requires finer-scale data and analysis.”

Discussion: Need to elaborate/ name what are those “different stressors”.

Changed. We have elaborated (L 232-236):

“Further, our analysis revealed little agreement between climate-smart prioritisations of landward and seaward edges of mangroves because of the different effects of some of the stressors involved (e.g., subsidence and sea-level rise have a positive effect on the landward edge but a negative effect on the seaward edge, and erosion only has a negative effect on the seaward edge)¹⁶. This might also be the case for other coastal habitats²⁴.”

Citations: need to provide citations for the “, the IUCN species distribution maps“.

Thanks. Changed. We now provide the reference (L 310-311):

“Our analysis has several caveats. First, the IUCN species distribution maps provide only broad distribution ranges¹⁹.”

References

1. Plumptre, A. *et al.* Strengths and complementarity of systematic conservation planning and Key Biodiversity Area approaches for spatial planning. *Conserv. Biol.* **39**, e14400 (2025).
2. Hanson, J. O. *et al.* Global conservation of species’ niches. *Nature* **580**, 232–234 (2020).
3. Claes, J. *et al.* *Valuing Nature Conservation*. (McKinsey & Company, 2022).
4. Convention on Biological Diversity. *Kunming-Montreal Global Biodiversity Framework*. (2022).
5. Cardinale, B. J. *et al.* Biodiversity loss and its impact on humanity. *Nature* **486**, 59–67 (2012).
6. Keck, F. *et al.* The global human impact on biodiversity. *Nature* **641**, 395–400 (2025).
7. Maxwell, S. L. *et al.* Area-based conservation in the twenty-first century. *Nature* **586**, 217–227 (2020).
8. Pressey, R. L., Mills, M., Weeks, R. & Day, J. C. The plan of the day: Managing the dynamic transition from regional conservation designs to local conservation actions. *Biol. Conserv.* **166**, 155–169 (2013).
9. Goué, A. M. & Kana, R. The Sangha Trinational: An example of cross-border biodiversity management in Central Africa. in *Managing Transnational UNESCO World Heritage*

- sites in Africa* (eds. Houehounha, D. & Moukala, E.) 69–81 (Springer International Publishing, Cham, 2023). doi:10.1007/978-3-030-80910-2_7.
10. Stoldt, M., Göttert, T., Mann, C. & Zeller, U. Transfrontier Conservation Areas and Human-Wildlife Conflict: The Case of the Namibian Component of the Kavango-Zambezi (KAZA) TFCA. *Sci. Rep.* **10**, 7964 (2020).
 11. Enright, S. R., Meneses-Orellana, R. & Keith, I. The Eastern Tropical Pacific Marine Corridor (CMAR): The Emergence of a Voluntary Regional Cooperation Mechanism for the Conservation and Sustainable Use of Marine Biodiversity Within a Fragmented Regional Ocean Governance Landscape. *Front. Mar. Sci.* **8**, (2021).
 12. Gnansounou, S. C. *et al.* The co-management approach has positive impacts on mangrove conservation: evidence from the mono transboundary biosphere reserve (Togo-Benin), West Africa. *Wetl. Ecol. Manag.* **30**, 1245–1259 (2022).
 13. Mungai, F. *et al.* Mangrove cover and cover change analysis in the transboundary area of Kenya and Tanzania during 1986–2016. *J. Indian Ocean Reg.* **15**, 157–176 (2019).
 14. Tuda, A. O., Kark, S. & Newton, A. Exploring the prospects for adaptive governance in marine transboundary conservation in East Africa. *Mar. Policy* **104**, 75–84 (2019).
 15. Dabalà, A. *et al.* Priority areas to protect mangroves and maximise ecosystem services. *Nat. Commun.* **14**, 5863 (2023).
 16. Buelow, C. A. *et al.* *Uncertainties in Forecasts of Ecosystem Persistence under Climate Change.* (2024). doi:10.21203/rs.3.rs-3708020/v1.
 17. Bunting, P. *et al.* Global Mangrove Extent Change 1996–2020: Global Mangrove Watch Version 3.0. *Remote Sens.* **14**, 3657 (2022).
 18. Buenafe, K. C. V. *et al.* A metric-based framework for climate-smart conservation planning. *Ecol. Appl.* **33**, e2852 (2023).
 19. IUCN. The IUCN Red List of Threatened Species. Version 2022-12. <https://www.iucnredlist.org/resources/spatial-data-download> (2022).
 20. Beger, M. *et al.* Demystifying ecological connectivity for actionable spatial conservation planning. *Trends Ecol. Evol.* **37**, 1079–1091 (2022).

21. Balbar, A. C. & Metaxas, A. The current application of ecological connectivity in the design of marine protected areas. *Glob. Ecol. Conserv.* **17**, e00569 (2019).
22. O'Regan, S. M., Archer, S. K., Friesen, S. K. & Hunter, K. L. A Global Assessment of Climate Change Adaptation in Marine Protected Area Management Plans. *Front. Mar. Sci.* **8**, 711085 (2021).
23. Adams, V. M. Costs in conservation: Common costly mistakes and how to avoid them. *PLOS Biol.* **22**, e3002676 (2024).
24. Friess, D. A., Adame, M. F., Adams, J. B. & Lovelock, C. E. Mangrove forests under climate change in a 2°C world. *WIREs Clim. Change* **13**, e792 (2022).
25. Krauss, K. W. & Osland, M. J. Tropical cyclones and the organization of mangrove forests: a review. *Ann. Bot.* mcz161 (2019) doi:10.1093/aob/mcz161.
26. Duke, N. C., Hutley, L. B., Mackenzie, J. R. & Burrows, D. Processes and Factors Driving Change in Mangrove Forests: An Evaluation Based on the Mass Dieback Event in Australia's Gulf of Carpentaria. in *Ecosystem Collapse and Climate Change* (eds. Canadell, J. G. & Jackson, R. B.) 221–264 (Springer International Publishing, Cham, 2021). doi:10.1007/978-3-030-71330-0_9.
27. Machava-António, V. C. E. *et al.* Massive mangrove dieback due to extreme weather impact - case of Maputo River Estuary, Mozambique. *Reg. Stud. Mar. Sci.* **78**, 103770 (2024).
28. IPCC. *The Ocean and Cryosphere in a Changing Climate: Special Report of the Intergovernmental Panel on Climate Change.* (Cambridge University Press, 2022). doi:10.1017/9781009157964.
29. Adame, M. F. *et al.* Future carbon emissions from global mangrove forest loss. *Glob. Change Biol.* **27**, 2856–2866 (2021).
30. Temmerman, S. *et al.* Ecosystem-based coastal defence in the face of global change. *Nature* **504**, 79–83 (2013).
31. Kelleway, J. J. *et al.* Review of the ecosystem service implications of mangrove encroachment into salt marshes. *Glob. Change Biol.* **23**, 3967–3983 (2017).

32. Rowland, P. I., Hagger, V. & Lovelock, C. E. Opportunities for blue carbon restoration projects in degraded agricultural land of the coastal zone in Queensland, Australia. *Reg. Environ. Change* **23**, 42 (2023).
33. Simard, M. *et al.* A New Global Mangrove Height Map with a 12 meter spatial resolution. *Sci. Data* **12**, 15 (2025).
34. Sunkur, R., Kantamaneni, K., Bokhoree, C., Rathnayake, U. & Fernando, M. Mangrove mapping and monitoring using remote sensing techniques towards climate change resilience. *Sci. Rep.* **14**, 6949 (2024).

REVIEWER COMMENTS

Reviewer #1 (Remarks to the Author):

Thank you for responding to my initial comments. I am satisfied with the revision.

Thank you, we are glad that you are satisfied with the revision. Thank you for the suggestions; they significantly improved the manuscript.

Reviewer #2 (Remarks to the Author):

I am generally satisfied with the responses to my initial comments. The authors have provided greater inclusion of geographical areas and clarity, which has improved the global appeal of the results. The figures are concise and informative and communicate greatly the results.

Thank you, we are happy that we were able to adequately reply to your comments. We believe they helped improve the manuscript.

However, I still think that there is much uncertainty in the results, evident by the way that the changes in the prioritisation of geographical area and species richness can change the results.

For example, the topic (and in the abstract) moves from "... a small increase ..." as this "Safeguarding climate-resilient mangroves requires a small increase in the global protected area" to a "... a moderate increase" as this "Safeguarding climate-resilient mangroves requires only a moderate expansion in the global protected area": This suggest that the main result has changed following the revision; why this change, any explanation for this? What scale was used to define "+7.3%" as "relatively moderate" expansion in protected area"?

The results did not change with the revision; we only ran additional sensitivity analyses. What did change was the replacement of the word "small" with "moderate," in response to a reviewer's request. We acknowledge that "moderate" is somewhat subjective, but there is no standardised scale for such comparisons. We have now removed the term "moderate" where possible. In cases where it was providing some context, we added the corresponding percentage in brackets. In the title, where we refer to more general findings, as highlighted by the results of our sensitivity analysis, we did not include any value.

From this change, it is clear that changes in the prioritisation selection for climate-naïve and climate smart species and geographical areas influences the results on the area of mangrove required to improve climate resilience of mangroves (ecosystems). Following this line of thought, I would suggest that the authors add a paragraph on the limitation of the methodology used (if any).

Changed. We included a sentence in the limitations paragraph in the discussion (L342-344):

"Seventh, additional tests confirm that the identification of specific conservation priorities was sensitive to input uncertainties. Nonetheless, our overall conclusions proved robust."

"Moderate increase" is used in the abstract while moderate expansion is used in the title; both have been used interchangeably, but do the words mean the same in the context and scope of this study? Do the authors use "increases" to mean increase in the area cover of PAs by creating new PAs and expansion to mean a growth in the size or areal extent of existing PAs? Some clarity my might help.

Changed. We now always use increase.

This minor comments do not undermine the quality of the work and its timely importance in conservation practice and decision making at larger spatial scales. I admire very much the inclusion of the sentence that the model is not applicable at local scale decision-making on mangrove conservation, this is very important to note.

Thank you!

Reviewer #3 (Remarks to the Author):

Authors significantly revised the manuscript.

All of my comments have been addressed.

Thank you. We appreciated your suggestions; they improved the manuscript.

Reviewer #3 (Remarks on code availability):

Technical Problem of my computer.

Reviewer #4 (Remarks to the Author):

Authors have sufficiently addressed the review points shown by me. I have no further suggestions on the revised manuscript.

Thank you. We appreciated your help with this work. Your suggestions did improve the manuscript.

Reviewer #4 (Remarks on code availability):

Authors have sufficiently addressed the review points shown by me. I have no further suggestions on the revised manuscript.